# Graph-based Confidence Calibration for Large Language Models

**Yukun Li**                                                                                         *yukun.li@tufts.edu*
*Department of Computer Science*
*Tufts University*

**Sijia Wang**                                                                                       *sijiawang@vt.edu*
*Department of Computer Science*
*Virginia Tech*

**Lifu Huang**                                                                                       *lfuhuang@ucdavis.edu*
*Department of Computer Science*
*Virginia Tech, University of California, Davis*

**Li-Ping Liu**                                                                                      *liping.liu@tufts.edu*
*Department of Computer Science*
*Tufts University*

**Reviewed on OpenReview:** *https://openreview.net/forum?id=BDPvuD5FTg*

## Abstract

Reliable confidence estimation is essential for enhancing the trustworthiness of large language models (LLMs), especially in high-stakes scenarios. Despite its importance, accurately estimating confidence in LLM responses remains a significant challenge. In this work, we propose using an auxiliary learning model to assess response correctness based on the self-consistency of multiple outputs generated by the LLM. Our method builds a consistency graph to represent the agreement among multiple responses and uses a graph neural network (GNN) to estimate the likelihood that each response is correct. Experiments demonstrate that this method has strong calibration performance on various benchmark datasets and generalizes well to out-of-domain cases.

## 1 Introduction

In recent years, large language models (LLMs) have demonstrated remarkable capabilities across various natural language processing tasks such as question answering (Wei et al., 2022; Shen et al., 2023; Zheng et al., 2023; Qin et al., 2023; Singhal et al., 2023), text summarization (Tang et al., 2023; Deroy et al., 2023; Tam et al., 2023; Roit et al., 2023), and even creative writing (Gómez-Rodríguez & Williams, 2023; Wang et al., 2024; Deng et al., 2024). Despite their impressive performance, LLMs often give incorrect responses in question-answering tasks. One particularly important challenge lies in calibrating the confidence levels of LLM-generated responses (Kuhn et al., 2022; Ulmer et al., 2022; Van Landeghem et al., 2022; Vazhentsev et al., 2023; Ulmer et al., 2024). Accurate confidence estimation is vital for deploying LLMs in the real world, as it enables users to gauge the reliability of the model's predictions and make informed decisions accordingly. On the contrary, miscalibrated confidence may lead to over-reliance on incorrect responses or unnecessary skepticism toward the correct ones. For example, a misleading response may steer a patient in the harmful direction when making health decisions; it may also cause an investor to make impulsive financial choices.

In this work, we focus on calibrating LLMs' confidence to better reflect the correctness of their responses. This task is challenging in several aspects. First, due to LLMs' superior ability to generate text, mistakes in their response often occur at the semantic level, making them hard to detect even for humans. There are methods using an auxiliary Language Model (e.g., DeBERTa (He et al., 2020)) to verify whether the LLM's response appropriately answers the question (Ulmer et al., 2024). Since the LLM is supposed to be much stronger than the LM, the LLM should be able to avoid most mistakes that can be detected by an LM; so this type of method may omit a significant fraction of wrong answers. Second, it is hard to detect mistakes from the LLM's internal working mechanism. Because the LLM uses many hidden layers to process the information, it is hard to discern the signal from a small number of hidden units. Even if this is possible, it is not easy to apply this type of method to black-box LLMs.

Recently, there has been some progress in quantifying the model's confidence in its own responses through consistency among the outputs generated by the model itself (Chen & Mueller, 2023; Lin et al., 2024). These approaches demonstrate a strong correlation between an LLM's self-consistency and the actual correctness of its responses. However, because these methods depend on hand-crafted features, they often fail to accurately calibrate confidence, resulting in a mismatch between predicted confidence levels and the true accuracy of the answers. This raises an important research question: can we improve confidence calibration by *learning* from patterns of consistency across the LLM's own responses?

In this work, we propose an auxiliary learning model to improve confidence calibration. We begin by constructing similarity graphs from the LLM's multiple responses to the same questions, where graph edges represent the degree of agreement between responses. We then train a separate calibration model using graph neural networks (GNNs) to predict the correctness of each response. The key insight is that consistency among responses carries strong signals of correctness – for instance, a response that aligns well with many others is more likely to be accurate. Importantly, our model operates solely on response consistency and does not analyze the actual language content. This work focuses on practical scenarios where correctness reflects alignment with the training data, and does not address the case where training data itself contains consistent but incorrect information. The latter case is a more challenging scenario investigated by ongoing research (Biester et al., 2024; Shi et al., 2023; Krishnan & Wu, 2019).

We further investigate the problem of transferring a calibration model across different question domains, which is crucial when target domains lack sufficient training data. Despite its importance, this problem has received limited attention in the literature. Our study demonstrates that the proposed auxiliary calibration model can generalize to new domains with minimal performance degradation. This generalization is from the observation that self-consistency can serve as a broadly applicable signal for confidence, enabling the learning model that relies solely on self-consistency to achieve strong transfer performance.

We evaluate the performance of the proposed method with an extensive empirical study that includes four datasets from different question domains. Empirical results show that our method achieves strong performance. Besides the improved calibration performance, our model enhances the ranking of an LLM's responses. The study has also tested our model and competing models in out-of-domain settings. The results show that the proposed method shows robust performance when generalizing to new domains.

In summary, our main contributions are:

- **A learning-based GNN framework:** We propose a learning-based framework leveraging GNNs to calibrate confidence values of LLMs' responses.

- **Enhanced calibration performance:** We conduct an extensive empirical study to evaluate the proposed method and show that it substantially outperforms recent methods in confidence calibration across several widely used benchmark datasets.

- **Improved out-of-domain generalizability:** We investigate the scenario of out-of-domain (OOD) confidence calibration and show the superior performance of the proposed method in this setting.

## 2 Related Work

Due to the urgent need to improve the reliability of LLMs, confidence estimation and calibration for these models have become active areas of research. Existing research in LLM uncertainty quantification can be summarized into two main categories: uncertainty quantification and confidence calibration (Geng et al., 2023). Confidence estimation for short responses (e.g., for multi-choice or yes-no questions) is generally less complicated than for long responses (Ye et al., 2024). For a brief response, the LLM's output logits are informative about its confidence; the easy comparisons of responses to the true answer facilitate both calibration and evaluation. Confidence estimation for long responses cannot simply depend on LLM's output logits (Duan et al., 2023; Bakman et al., 2024) because the logits indicate more about the probability of text and less about the semantics behind it. There are also methods using the internal state of an LLM (Ren et al., 2022; Beigi et al., 2024), but it is not always available to have such information about the LLM interface.

Another approach is to check the LLM's consistency in its responses. Kotelanski et al. (2023) demonstrate that repeated sampling and consistency checks across multiple outputs can serve as reliable proxies for model confidence. Manakul et al. (2023) generate multiple responses from the LLM and check the consistency between responses using various methods, including querying the LLM. Chen & Mueller (2023) combine the consistency between responses and the LLM's self-reflection certainty to quantify the uncertainty. Kuhn et al. (2022) consider confidence from semantic equivalence and proposes a method based on clustering of responses. Lin et al. (2024) organize responses in a graph with their pairwise semantic similarity and then extract graph statistics for confidence estimation. Zhang et al. (2024) examine methods of comparing responses via entailment and contradiction relationships. These studies highlight the importance of semantic consistency in ranking an LLM's responses. However, manually designed features are limited in their ability to capture the full extent of self-consistency among LLM responses, leading to poor calibration performance.

To better calibrate the confidence estimation, some methods directly use correctness labels in their calibration procedures. Mielke et al. (2022) train a calibrator to predict the correctness of a response for a given question. With a similar idea, Ulmer et al. (2024) train a language model (e.g., DeBERTa) based on question-response pairs to predict the probability of responses' correctness. Based on SelfCheckGPT (Manakul et al., 2023) and JAFC (Tian et al., 2023), Chen et al. (2024) train supervised models to reduce grouping losses and improve the confidence estimation. The method by Liu et al. (2024) uses an LLM's latent representations to predict the correctness of responses. Detommaso et al. (2024) use the "multicalibration" technique to calibrate the probability of correctness. Fadeeva et al. (2023) offer a detailed comparative study of various confidence estimation methods, providing empirical evidence on their effectiveness across different tasks. However, these studies have not sufficiently exploited response consistency to predict the probabilities of the responses being correct.

## 3 Method

Our ultimate goal is to quantify the probability of the correctness of a response from an LLM. Since the LLM can give a correct answer with different phrases, we need to consider the probability that the response is semantically correct.

**Background:** The formulation of semantic equivalence (Kuhn et al., 2022) provides a framework for our analysis. Let $\mathcal{R}$ be the space of all possible responses. Given a question $q$, the space $\mathcal{R}$ is divided into a set $\mathcal{C}_q$ of semantic classes: $\mathcal{R} = \cup_{C \in \mathcal{C}_q} C$ and $C' \cap C = \emptyset$ for any two different semantic classes $C, C' \in \mathcal{C}_q$. Two responses $r_1, r_2 \in C$ in the same equivalent class are considered as the same semantic response: if one is the correct answer, the other is correct as well. Under an LLM, the semantic response $C$ has probability:

$$p(C|q) = \sum_{r \in C} p(r|q). \tag{1}$$

Here $p(r|q)$ is the probability of a single response from the LLM. We also say that $p(C|q)$ is the LLM's confidence in the semantic response $C$.

However, it is non-trivial to define the equivalent class, and we will discuss the approximation later. To estimate $p(C|q)$, one approach is through semantic similarities between response samples of an LLM for the same question $q$. Let $(r_1, \ldots, r_n)$ be $n$ responses from the same question $q$, and they form $k$ clusters $\tilde{\mathcal{C}}_q = \{\tilde{C}_1, \ldots, \tilde{C}_k\}$ by their semantic similarity. We can use natural language inference (NLI) systems to predict the relationships (e.g., entailment and contradiction) between responses and derive their similarity.

We assume that each cluster $\tilde{C}$ is from a different semantic class $C$, then $p(C|q)$ can be approximated by

$$p(C|q) \approx \frac{|\tilde{C}|}{n}. \tag{2}$$

From the cluster probabilities, the uncertainty of the LLM on the question $q$ is estimated as the entropy of the empirical distribution over clusters (Kuhn et al., 2022), and the confidence of a response $r_i \in C$ is estimated as $|\tilde{C}|/n$ (assuming similarity values are binary) (Lin et al., 2024).

Now, we depart from the setup of semantic classes and consider the correctness of responses. Let $C^*$ be the correct semantic answer to question $q$. Without knowing $C^*$, a common assumption is that the model's confidence $p(C|q)$ in the semantic response $C$ reflects the correctness. For example, $p(C|q)$ *is* approximately the probability of correctness:

$$p(\tilde{C}_{k'} \subseteq C^*) \approx \frac{|\tilde{C}_{k'}|}{n}. \tag{3}$$

With this assumption, the more certain the model is about a semantic response, the more likely the response is correct. Conversely, a wide variation in the LLM's responses indicates low confidence in all responses $r_i$ and low accuracy. This pattern is also found in previous studies (Kuhn et al., 2022; Lin et al., 2024).

While a positive correlation exists between the LLM's confidence and the probability of correctness, *the two quantities are unlikely to have the equal relationship shown in* (3). Therefore, we need further calibration to estimate the probability of correctness.

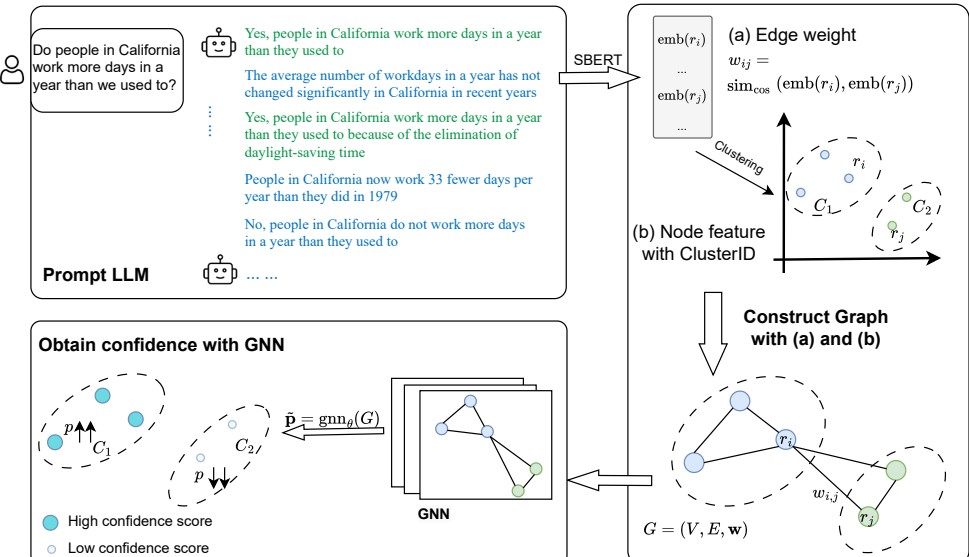

Figure 1: **The overall framework of our confidence calibration model**. Given an input question, our approach first generates multiple responses from the LLM and constructs a similarity-weighted graph based on these responses. This graph serves as the input for the GNN model, which calibrates the confidence of the LLM responses. In the weighted graph, the edge weight $w_{ij}$ is defined as $\mathrm{sim}_{\cos}(\mathrm{emb}(r_i), \mathrm{emb}(r_j))$, where $i, j = 1, \ldots, n$. A higher weight indicates greater similarity between the responses. We also use cluster memberships as the node features to enhance the performance.

### 3.1 Confidence calibration as graph learning problem

Now, we set a supervised learning problem and train a model to calibrate the confidence of the correctness of responses. We first consider the correctness labels of the LLM's responses. In the supervised setting, we have a ground-truth answer $a$ to the question $q$. Then we use $a$ to assign correctness labels to sampled responses $\{r_1, r_2, ..., r_n\}$ for the same question $q$.

In our work, we have two approaches to label these responses. In the first approach, we use the ROUGE similarity. Specifically, we compute the ROUGE similarity $\text{sim}_R(r_i, a)$ between a sampled response and the correct answer to decide the correctness label.

$$y_i = \mathbb{1}[\text{sim}_R(a, r_i) \geq \tau], \ i = 1, \ldots, n. \tag{4}$$

Here $\mathbb{1}[\cdot]$ is one if the condition is true or 0 otherwise. As shown in previous studies (Lin & Och, 2004) and our own study, the ROUGE metric is reasonably accurate in measuring semantic similarity between short sentences. We follow the previous work, and set $\tau = 0.3$ (Kuhn et al., 2022).

In the second method, we utilize the LLM to generate correctness labels. Specifically, we provide the question $q$ and the standard answer $a$ as the context, then ask whether the response $r_i$ answers the question $q$. The response from the LLM is then used as the label for $r_i$. We denote the procedure as

$$y_i = \text{llm}_y(q, a, r_i) \tag{5}$$

We provide the prompt for labeling in the Appendix F.

Both methods are automatic and can scale up to large datasets. To guaranteed the labeling quality, we also include a relatively small set of manual labels in our study. For each question, a human labeler inspects the question $q$ and the true answer $a$, and then assign the correctness label $y_i$ to a response $r_i$.

We then consider the input to the calibration model. We form a similar graph $G$ over responses to encode information about their consistency. The graph contains the clustering structure of responses and likely further useful information to predict the correctness of responses. The graph $G = (V, E, \mathbf{w})$ is a fully connected graph, with the node set $V$ consisting of $n$ responses and the edge weight $w_{ij}$ being the similarity between the pair of responses $(r_i, r_j)$. We compute the similarity from the two responses' embeddings. In particular, we first use the Sentence-BERT model (Reimers & Gurevych, 2019) to compute the two responses' vector representations and then compute the cosine similarity

$$w_{ij} = \text{sim}_{\cos}(\text{emb}(r_i), \text{emb}(r_j)), \ \ i, j = 1, \ldots, n. \tag{6}$$

Here, $\text{emb}(\cdot)$ represents the embedding function.

Then, we treat the problem as a node classification problem (Xiao et al., 2022). In particular, we run a GNN $\text{gnn}(\cdot)$ to predict the probability of each response being correct

$$\tilde{\mathbf{p}} = \text{gnn}_\theta(G). \tag{7}$$

Here $\tilde{\mathbf{p}} \in [0, 1]^n$ contains the probabilities for $n$ responses being correct.

To provide clustering information to the GNN, we first run the K-means clustering algorithm on the responses' embeddings and assign cluster ids from 0 to $K - 1$ based on the order from largest to smallest (ties are randomly broken). Then, we feed each response's cluster membership as a one-hot feature input to the GNN. Therefore, the GNN's predictions are purely based on the relationships between responses in semantic space. We choose NOT to feed in the embedding vectors of responses to avoid the GNN's dependency on textual information. This helps the GNN to generalize to questions from different domains. The overall framework is shown in Fig 1.

The main purpose of the learning model is to calibrate $\tilde{\mathbf{p}}$. One approach is to minimize the cross-entropy loss of $\tilde{\mathbf{p}}$ against correctness labels. The loss computed from the question $q$ is

$$\ell_q = -\sum_{i=1}^n y_i \log \tilde{p}_i + (1 - y_i) \log(1 - \tilde{p}_i) \tag{8}$$

Note that the loss is consistent marginally since the loss is minimized when $\tilde{p}_i = p(y_i|G)$. An alternative approach is to minimize the squared error $(y_i - \tilde{p}_i)$, from which we get similar performances. We choose the cross-entropy loss in our work. A further consideration is to explicitly consider the similarity between $\tilde{p}_i$ and $\tilde{p}_j$ given the response similarity $w_{ij}$. We leave such exploration to the future.

## 3.2 Improve the estimation through multiple prompts

It is well known that the syntactic form of a question influences responses and introduces additional variance. The variance of question forms from users may pass to the variance of responses and cause even diverse responses due to the lack of confidence. To account for question variances in real applications, we analyze the LLM's responses to multiple prompts derived from the same question. These responses are treated as answers to the same semantic question. We then apply the same method as before to predict the correctness of each response.

In particular, we rephrase the original question $q$ into $k$ different forms $\{q_1, ..., q_k\}$ while maintaining the original sentence's semantic meaning. We employ a multiple rephrased questions strategy for answer sampling. Specifically, we prompt the GPT-4 to give $k$ different but with the same meaning rephrased questions for the given question $q$. Then, we sample $n/k$ responses from the LLM for each rephrased question and still get a total of $n$ responses, from which the confidence calibration is the same as we have described above. For questions about which the LLM is less certain, the model is more likely to produce diverse responses. In this scenario, confidence calibration is more accurate because the model's uncertainty becomes more apparent.

# 4 Experiments

We evaluate the calibration performance of our proposed framework by comparing it against baseline methods. We also test their performances in the OOD setting, where their calibration parameters are decided on one dataset, but they are tested on a different dataset. We further analyze the quality of automatic labels and the sensitivities of our method under different hypter-parameter settings.

## 4.1 Dataset and experiment setup

**Dataset**: We conduct experiments on four datasets: (1) CoQA (Reddy et al., 2019), an open-book conversational question answering task; (2) TriviaQA (Joshi et al., 2017), a commonsense QA task. (3) TruthfulQA (Lin et al., 2022a), a comparably more challenging dataset for factual QA tasks. and (4) HotpotQA, a question answering dataset that requires models to find and combine information from multiple passages to answer complex questions. We repeat the experiments 10 times, each with a different train/validation split and test the performance on the test set.

**Baselines**: We compare our methods with the following baselines. Length-normalized sequence likelihoods (**Seq. likelihood**) (Malinin & Gales, 2021; Kuhn et al., 2022) is a standard measure for confidence. This method calculates the likelihood of each sequence and normalizes it by the length of the sequence to provide a fair comparison between different lengths of sequences. **Platt scaling** (Platt, 1999), a variant of the sequence likelihood baseline, applies Platt scaling to the raw likelihoods. **GraphSpectral** (Lin et al., 2024) uses the graph theory to estimate the confidence. Then we also include post-hoc uncertainty calibration, **GraphSpectral+Iso** and **GraphSpectral+Platt** into the baseline methods. **Self-check GPT**(Manakul et al., 2023) checked the consistency between responses querying the LLM. **Verbalized Uncertainty** (Lin et al., 2022b; Tian et al., 2023; Xiong et al., 2024) generates verbal statements about the model's confidence in its predictions. Verbalized Qual maps the confidence percent (Verbalized %) into numerical values. **APRICOT** (Ulmer et al., 2024), a supervised method, fine-tunes the Deberta language model to predict confidence scores for LLM outputs. Furthermore, we also include the baseline of applying two post-hoc uncertainty calibration methods, **APRICOT+Iso** and **APRICOT+Platt**, to adjust the confidence scores obtained by Apricot. We performed all the baseline experiments utilizing the open-source codebase and used the default hyperparameters.

**Graph construction:** For each question, we generate 30 candidate answers through LLM prompting. Each generated response is then processed using the Sentence-BERT (SBERT) model (Reimers & Gurevych, 2019) to obtain the answer's high-dimensional embeddings. To quantify the semantic alignment between all responses, we compute the cosine similarity between every pair of answer embeddings. These scores are then utilized as edge weights in our similarity graph, where each node represents an individual answer, and the edges weights signify the degree of semantic relation between them.

**Model hyper-parameters**: To ensure our model can capture complex and abstract features at each layer, our model comprises three GNN layers, with embedding dimensions of 256, 512, and 1024 for the first, second, and third layers, respectively. The initial learning rate was set to $10^{-4}$. If the validation loss did not show improvement over ten consecutive epochs, the learning rate was reduced by a factor of 0.9. The optimization was performed using the Adam optimizer, configured with hyperparameters $\beta_1 = 0.9$ and $\beta_2 = 0.98$. The batch size was 16.

**LLMs**: We assess our confidence calibration method on two LLMs with good performance: `Llama3-8B` (Llama3)(Meta, 2024), and `Vicuna-7b-v1.5` (Vicuna) (Zheng et al., 2024).

**Labeling the data**: To obtain the correctness label for CoQA and TriviaQA datasets, we followed previous work (Kuhn et al., 2022) and used the ROUGE-L metric for labeling. For the TruthfulQA dataset, given its focus on factual correctness and longer answers, we employed GPT4 (Potsaweel, 2024; Liu et al., 2023; Badshah & Sajjad, 2024) to generate the labels. In addition, we conducted experiments on a smaller subset of data using human-annotated labels to validate the reliability of our automatic labeling process. Details of this manual annotation study are provided in Section4.4.

**Evaluation metrics**: The evaluation metrics include Expectation Calibration Error (ECE), Brier Score, and AUROC. Specifically, (1) **ECE** quantifies the consistency between the prediction error and the uncertainty of the prediction. An ideal calibration curve should exhibit a lower ECE. It measures the consistency between the prediction error and the confidence of the prediction. Specifically, the confidence interval is grouped into fixed bins, and the average of the difference between the confidence and error in each bin is compared. Formally, ECE is calculated as $ECE = \sum_{b=1}^{B} \frac{n_b}{N} |acc(b) - conf(b)|$, where $n_b$ is the number of predictions in bin $b$, $N$ is the total number of data points and $acc(b)$ and $conf(b)$ are the accuracy and confidence of bin b, respectively. (2) **Brier Score** (Brier, 1950), which is the mean squared difference between predicted probabilities and the actual binary results. Lower Brier Scores indicate better performance. (3) **AUROC** to indicate the models' discriminatory ability.

Further experimental configurations and prompting strategy are provided in the Appendix. The experiments are conducted on NVIDIA A100 GPUs with 80GB of memory.

## 4.2 Experiment results

For the Llama3 model, the confidence calibration performance on TriviaQA is shown in Table 1. For the TriviaQA dataset, it can be observed that the likelihood-based method performs poorly on the calibration error (ECE and Brier Score) and AUROC due to unreliable model prediction probability (Zhang et al., 2024). Platt scaling improves the ECE post-calibration and enhances the model's discriminative ability, resulting in higher AUROC results. However, this method cannot capture the semantic equivalence among answers, leading to sub-optimal performance. The Verbalized and Verbalized Qual prompts LLM to output confidence for their answers, improving AUROC by $3 - 5\%$ compared with the likelihood baseline. However, it faces the overconfidence issue; thus, the calibration errors are still high. The GraphSpectral method can produce good confidence estimations, but its calibration performance is poor. Even with the addition of techniques such as Isotonic Calibration or Platt Scaling, this issue can only be partially mitigated. The auxiliary DeBERTa method combines the LLM outputs, Chain-of-Thoughts (CoT) outputs, and verbalized confidence to fine-tune the DeBERTa model for predicting confidence. Our method captures the prediction confidence based on the graph structure of LLM's responses in semantic space and achieves better ECE results. The ECE is reduced from 0.07 to 0.03 and improves the AUROC from 0.72 to 0.86 compared with the baseline calibration methods. The experiment results on TruthfulQA, HotpotQA and CoQA for the Llama3 model are shown in Table 1. These results show a similar trend, with our model achieving superior performance in confidence calibration compared to the baseline methods.

Table 1: Comparison of confidence calibration performance on TriviaQA, CoQA, TruthfulQA and HotpotQA dataset for Llama3

| Method | TriviaQA | | | CoQA | | | TruthfulQA | | | HotpotQA | | |
|---|---|---|---|---|---|---|---|---|---|---|---|---|
| | Brier↓ | AUROC↑ | ECE↓ | Brier↓ | AUROC↑ | ECE↓ | Brier↓ | AUROC↑ | ECE↓ | Brier↓ | AUROC↑ | ECE↓ |
| GraphSpectral (GS) | .223 ± .002 | .842 ± .002 | .0762 ± .007 | .193 ± .001 | .762 ± .008 | .110 ± .019 | .332 ± .002 | .667 ± .012 | .239 ± .019 | .172 ± .017 | .783 ± .006 | .097 ± .014 |
| GS + Iso | .167 ± .011 | .842 ± .002 | .058 ± .002 | .162 ± .008 | .762 ± .008 | .054 ± .002 | .191 ± .015 | .667 ± .012 | .088± .007 | .163 ± .012 | .783 ± .006 | .087 ± .021 |
| GS + Platt | .165 ± .012 | .842 ± .002 | .049 ± .002 | .161 ± .009 | .762 ± .008 | .042 ± .001 | .221 ± .013 | .667 ± .012 | .151± .008 | .160 ± .014 | .783 ± .006 | .177 ± .012 |
| Self-checkGPT | .332 ± .031 | .652 ± .020 | .187 ± .002 | .209 ± .020 | .633 ± .027 | .178 ± .010 | .362 ± .028 | .566 ± .028 | .353± .030 | .283 ± .014 | .673 ± .022 | .122 ± .030 |
| Seq. likelihood | .536± .015 | .591 ± .002 | .220 ± .002 | .382 ± .012 | .571 ± .028 | .173 ± .009 | .465 ± .008 | .582 ± .025 | .052 ± .009 | .463 ± .018 | .651 ± .002 | .105± .012 |
| Platt | .276 ± .006 | .591 ± .002 | .052 ± .002 | .258 ± .000 | .571 ± .0280 | .090 ± .009 | .271 ± .007 | .582 ± .025 | .053 ± .008 | .220 ± .0120 | .651 ± .002 | .142± .008 |
| Verbalized Qual | .322 ± .034 | .618 ± .002 | .142 ± .002 | .302 ± .021 | .681 ± .022 | .160 ± .007 | .320 ± .037 | .622 ± .016 | .140 ± .008 | .358 ± .012 | .652 ± .006 | .150 ± .029 |
| Verbalized % | .253 ± .021 | .663 ± .008 | .033 ± .002 | .423 ± .0120 | .662 ± .027 | .216 ± .002 | .540 ± .035 | .573 ± .029 | .331 ± .008 | .319 ± .014 | .672 ± .002 | .220± .023 |
| APRICOT | .145 ± .002 | .723 ± .003 | .074 ± .005 | .173 ± .006 | .751 ± .022 | .132 ± .006 | .201 ± .003 | .657 ± .034 | .062 ± .011 | .171 ± .014 | **.823 ± .011** | .081± .009 |
| APRICOT+Iso | .182 ± .012 | .723 ± .003 | .073 ± .004 | .171 ± .009 | .751 ± .022 | .097 ± .003 | .200 ± .003 | .657 ± .034 | .059 ± .011 | .180 ± .012 | **.823 ± .011** | .073± .002 |
| APRICOT+Platt | .173 ± .018 | .723 ± .003 | .042 ± .004 | .169 ± .012 | .751 ± .022 | .069 ± .008 | .230 ± .003 | .657 ± .034 | .056 ± .011 | .171 ± .018 | **.823 ± .011** | .071± .010 |
| Ours | **.136 ± .000** | **.864 ± .002** | .035 ± .004 | .124 ± .000 | .768 ± .009 | **.013 ± .003** | **.151 ± .003** | .732 ± .012 | **.037±.013** | **.142 ± .000** | .815 ± .002 | .023 ± .004 |
| Ours(Multi prompts) | .141 ± .002 | .853 ± .002 | .036 ± .008 | **.118 ± .000** | **.776 ± .012** | .015 ± .007 | .173 ± .003 | **.736 ± .007** | .039 ± .013 | **.142 ± .000** | .821 ± .002 | **.021 ± .006** |

Table 2: Comparison of confidence calibration performance on TriviaQA, CoQA, TruthfulQA and HotpotQA dataset for Vicuna

| Method | TriviaQA | | | CoQA | | | TruthfulQA | | | HotpotQA | | |
|---|---|---|---|---|---|---|---|---|---|---|---|---|
| | Brier↓ | AUROC↑ | ECE↓ | Brier↓ | AUROC↑ | ECE↓ | Brier↓ | AUROC↑ | ECE↓ | Brier↓ | AUROC↑ | ECE↓ |
| GraphSpectral (GS) | .196±.000 | .792±.006 | .112±.014 | .275±.004 | .696±.004 | .202±.011 | .286±.006 | .647±.009 | .226±.013 | .162±.076 | .673±.008 | .202±.009 |
| GS + Iso | .196±.000 | .792±.006 | .059±.008 | .245±.002 | .696±.004 | .037±.014 | .297±.008 | .647±.009 | .092±.004 | .165±.058 | .673±.008 | .085±.028 |
| GS + Platt | .172±.000 | .792±.006 | .067±.009 | .228±.002 | .696±.004 | .055±.028 | .307±.007 | .647±.009 | .183±.003 | .160±.049 | .673±.008 | .073±.049 |
| Self-checkGPT | .355±.001 | .640±.003 | .183±.014 | .221±.004 | .648±.009 | .192±.010 | .281±.012 | .552±.008 | .308±.017 | .295±.187 | .652±.187 | .370±.187 |
| Seq. likelihood | .485±.002 | .581±.002 | .420±.029 | .302±.012 | .688±.002 | .169±.090 | .325±.022 | .587±.010 | .205±.012 | .493±.220 | .630±.050 | .223±.220 |
| Platt | .342±.002 | .581±.002 | .255±.016 | .308±.015 | .688±.002 | .165±.008 | .288±.014 | .587±.010 | .181±.005 | .232±.050 | .630±.050 | .259±.050 |
| Verbalized Qual | .393±.002 | .631±.007 | .029±.014 | .455±.022 | .495±.004 | **.009±.001** | .471±.034 | .482±.060 | **.018±.005** | .220±.140 | .652±.140 | .142±.140 |
| Verbalized % | .402±.001 | .523±.005 | .383±.012 | .492±.025 | .539±.003 | .324±.029 | .580±.022 | .566±.009 | .387±.017 | .342±.033 | .683±.033 | .033±.033 |
| APRICOT | .196±.000 | .783±.006 | .068±.007 | .193±.004 | .742±.006 | .073±.009 | **.197±.007** | .769±.002 | .118±.005 | .152±.001 | .782±.022 | .074±.074 |
| APRICOT+Iso | .187±.000 | .783±.006 | .049±.004 | .193±.005 | .742±.006 | .064±.007 | **.197±.007** | .769±.002 | .092±.008 | .142±.001 | .782±.022 | .073±.073 |
| APRICOT+Platt | .186±.000 | .783±.006 | .052±.004 | .193±.005 | .742±.002 | .049±.004 | .204±.006 | .769±.002 | .085±.005 | .150±.001 | .782±.022 | .042±.042 |
| Ours | .169±.000 | **.816±.002** | .028±.004 | .184±.001 | .754±.004 | .032±.004 | .202±.003 | **.774±.001** | .059±.006 | .132±.000 | **.791±.002** | **.022±.002** |
| Ours(Multi prompts) | **.165±.000** | .815±.006 | **.025±.003** | **.168±.001** | **.763±.004** | .030±.006 | .202±.004 | .764±.001 | .063±.004 | **.131±.000** | .790±.003 | .025±.009 |

Furthermore, we also compare the confidence calibration performance for the Vicuna model on the TriviaQA, CoQA, TruthfulQA and HotpotQA datasets. The results are summarized in Table 2. Our model consistently improves the calibration error compared to the baseline methods. Both GraphSpectral and our method have a similar assumption that the consistency level between responses indicates the confidence levels of these responses. However, GraphSpectral uses simple graph statistics to measure the confidence level of responses and could not capture complex relationships between response patterns and confidence levels (e.g. patterns beyond clustering structures). As a comparison, by framing the problem as a learning problem, our method has better opportunities to discover such relationships and provides a better calibration performance. Self-Check GPT uses its own evaluation on whether the context supports the answers and heavily relies on the LLM model's capability to do self-reflection, which can also be hallucinated. Thus the generated confidence scores are not calibrated well with empirical accuracy.

We present the reliability diagrams for all methods on TriviaQA to better understand the model improvement. The reliability diagram is created by binning responses into 10 bins according to their confidence values. Then the frequency of correctness is also computed for each bin. With an ideal calibration, the confidence value in each bin should match the frequency of correctness. At the same time, a good calibration model

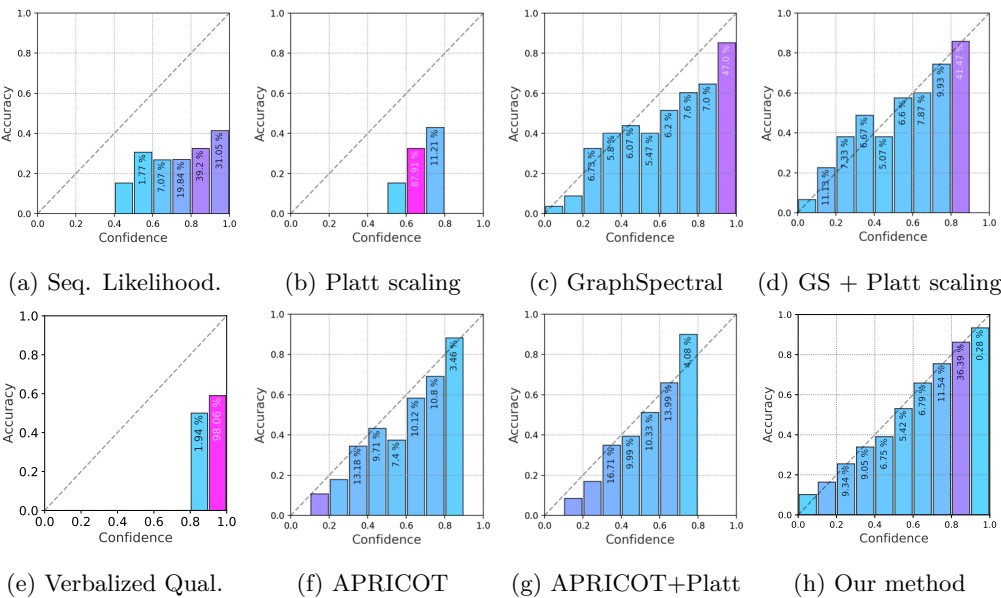

(a) Seq. Likelihood.   (b) Platt scaling   (c) GraphSpectral   (d) GS + Platt scaling

(e) Verbalized Qual.   (f) APRICOT   (g) APRICOT+Platt   (h) Our method

Figure 2: Reliability diagrams for different methods using 10 bins each for Vicuna on TriviaQA. The color, as well as the percentage number within each bar, indicates the proportion of total responses contained in each bin. Larger values are represented by colors closer to purple, and the height indicates the ratio of correct ones. We prefer a wide spread of responses in different bins (strong ability to differentiate responses) and bin heights along the diagonal line (accurate calibration). Our model outperforms others with a broader bin spread and better alignment with the diagonal for calibration accuracy.

should differentiate responses according to their confidence levels and give a wide distribution of confidence values in these bins.

The reliability diagram is shown in Fig. 2. (We also show other reliability diagrams for the different methods for Llamas on TriviaQA and CoQA in the Appendix E). The figure presents the reliability diagrams for different methods. In these diagrams, both the color intensity and the percentage numbers within each bar represent the proportion of total responses that fall into each respective bin. Specifically, larger proportions are depicted with colors closer to purple, while the height of each bar indicates the ratio of correct predictions within that bin. Our framework achieves a broad spread of responses across the bins, showing good differentiation capabilities; at the same time, the bar heights closely follow the diagonal line, indicating a good calibration performance. However, baseline methods cannot reach the same performance. The likelihood-based confidence methods exhibit significant overconfidence, indicating many responses rated with high confidence are actually wrong. The Auxiliary DeBERTa (APRICOT) method, which integrates LLM outputs, Chain-of-Thought (CoT) outputs, and verbalized confidence to train an auxiliary DeBERTa model, enhances the AUROC. However, it still experiences some overconfidence issues, potentially caused by the inherent overconfidence in the input verbalized confidence scores. Furthermore, the baseline methods' reliability diagrams revealed that this method frequently assigned high confidence scores to incorrect predictions, deviating markedly from the ideal calibration represented by the diagonal line. For example, the verbalized method's predictions in the highest confidence bins (80-90%) were significantly below the corresponding empirical accuracy, indicating a tendency to overestimate the certainty of its outputs.

### 4.3 Confidence calibration in the OOD setting

Domain shift poses significant challenges for deploying machine learning models in real-world scenarios where data variability is expected. To comprehensively assess the robustness and generalization capabilities of our proposed model compared to baseline methods, we conducted a series of out-of-domain (OOD) evaluations.

Table 3: Evaluation in the OOD setting. Models are trained on the TriviaQA from Llama3 responses and tested on out-of-domain datasets

| Dataset | Method | Brier | AUROC | ECE |
|---|---|---|---|---|
| Llama3 CoQA | GraphSpectral(w platt) | 0.17 | 0.72 | 0.095 |
| | Apricot | 0.24 | 0.59 | 0.154 |
| | Ours | **0.13** | **0.77** | **0.077** |
| Llama3 TruthfulQA | GraphSpectral(w platt) | 0.32 | 0.63 | 0.324 |
| | Apricot | 0.25 | 0.54 | 0.197 |
| | Ours | **0.23** | **0.66** | **0.16** |
| Vicuna TriviaQA | GraphSpectral(w platt) | 0.24 | 0.53 | 0.07 |
| | Apricot | 0.19 | 0.76 | 0.13 |
| | Ours | **0.17** | **0.81** | **0.07** |
| Vicuna CoQA | GraphSpectral(w platt) | 0.35 | 0.55 | 0.26 |
| | Apricot | 0.24 | 0.59 | **0.08** |
| | Ours | **0.22** | **0.73** | 0.10 |

**Experiment setup:** We evaluate the confidence calibration of different approaches under out-of-domain settings. We have two experiment configurations: **out-of-domain dataset OODD**, and **out-of-domain LLMs (OODL)**. For OODD, we train the confidence calibration model on TriviaQA from Llama3 responses and test it on CoQA Llama3 and TruthfulQA Llama3 answers. For OODL, we use the same training data from Llama3 but test the Vicuna model's responses on the TriviaQA and CoQA datasets. We compare our model with the Apricot and GraphSpectral (with Platt scaling) methods.

**Results and Analysis:** Table. 3 shows the OOD performance of the baseline methods. The OOD experiment results revealed that our model maintained a high level of performance across tested domains. Specifically, the model demonstrated consistent calibration, as evidenced by low ECE values and strong discriminative ability, reflected in high AUROC scores on in-domain and OOD datasets. For example, while the model achieved an ECE of 0.03 and an AUROC of 0.86 on TriviaQA (in-domain), it maintained an ECE of 0.077 and an AUROC of 0.77 on CoQA. Furthermore, the Brier scores across domains remained within acceptable ranges, demonstrating reliable probabilistic predictions even when faced with unfamiliar data distributions. The relatively small increase in ECE and a slight decrease in AUROC for OOD datasets suggest that while there is some degradation in performance, the model retains substantial robustness and accuracy. This is primarily because similarity graph patterns are highly invariant to the data distribution. Specifically, our model employs the consistency graph and the clustering feature that does not alter with data distribution shifts, enabling it to maintain stable performance across different datasets.

In contrast, Apricot typically relies on specific dataset features, which leads to poor performance in OOD scenarios. Furthermore, calibration methods like the Platt scaling can improve the confidence calibration in-domain, but their calibration effectiveness remains limited under domain shift scenarios. This is because this calibration technique mainly adjusts the output probabilities but does not fundamentally address the biases introduced by feature representation changes across distributions.

## 4.4 Checking the quality automatic labels

In our previous experiment, we used an automated method to assign the correctness labels to responses by comparing them to true answers with ROUGE-L scores. One question is whether these labels are reliable. In this experiment, we manually annotate the labels of a small set of LLM-generated responses and examine the accuracy of labels generated from ROUGE-L scores. We also use these manual labels to evaluate top-performing algorithms.

To label the data manually, a human labeler checks the true answer of a question and labels the correctness of a response generated by the LLM. Then, automatic labels are compared to manual labels to get the accuracy value. Limited by our resources, we label 600 questions for each of the three datasets, TriviaQA, CoQA, and HotpotQA. Table 4 provides a breakdown of the accuracy across different datasets. The results

Table 4: The accuracy of labels generated from ROUGE-L scores. The high accuracy indicates the reliability of automatic labels, as well as our evaluation above.

|  | TriviaQA | CoQA | HotpotQA |
|---|---|---|---|
| Accuracy of automatic labels | 0.96 | 0.92 | 0.89 |

Table 5: Evaluation with manual labels (Llama3)

| Dataset | Method | AUROC | ECE | Brier |
|---|---|---|---|---|
| TriviaQA | GS+Platt | 0.79 | 0.050 | 0.15 |
|  | Apricot+Platt | 0.77 | 0.059 | 0.13 |
|  | Ours | 0.81 | 0.037 | 0.12 |
| CoQA | GS+Platt | 0.74 | 0.056 | 0.12 |
|  | Apricot+Platt | 0.72 | 0.062 | 0.11 |
|  | Ours | 0.76 | 0.023 | 0.09 |
| HotpotQA | GS+Platt | 0.76 | 0.088 | 0.19 |
|  | Apricot+Platt | 0.78 | 0.052 | 0.18 |
|  | Ours | 0.81 | 0.030 | 0.15 |

Table 6: Evaluation with manual labels (Vicuna)

| Dataset | Method | AUROC | ECE | Brier |
|---|---|---|---|---|
| TriviaQA | GS+Platt | 0.78 | 0.053 | 0.18 |
|  | Apricot+Platt | 0.77 | 0.052 | 0.17 |
|  | Ours | 0.80 | 0.032 | 0.16 |
| CoQA | GS+Platt | 0.69 | 0.049 | 0.19 |
|  | Apricot+Platt | 0.72 | 0.057 | 0.15 |
|  | Ours | 0.77 | 0.032 | 0.12 |
| HotpotQA | GS+Platt | 0.68 | 0.065 | 0.13 |
|  | Apricot+Platt | 0.77 | 0.076 | 0.14 |
|  | Ours | 0.78 | 0.041 | 0.12 |

indicate that labels from ROUGE-L scores are fairly accurate: the lowest accuracy is 0.89. The results are also consistent with previous studies (Kuhn et al., 2022).

We further evaluated our model and two strong baselines (GraphSpectral+Platt and Apricot+Platt) with manual labels. As shown in Table 5 and Table 6, our model still outperforms the baselines, which is consistent with our previous evaluation results with automatic labels.

## 4.5 Sensitivity analysis

In this subsection, we conducted several sensitivity analyses of our model.

**Number of training samples** We conducted experiments to examine the relationship between performance and the amount of training data. Specifically, we tested our model performance on the Llama3 TriviaQA dataset and varied the training size from 100 to 4000. The results are displayed in Table 7. We observed that the model's performance does not drop significantly with the reduced training data. These experimental results indicate that the model performs well with limited data availability, demonstrating its applicability in real-world scenarios where only smaller datasets are available. We also tested the baseline performance, the results are shown in Appendix E.

**Hyperparameter sensitivity** We conduct the sensitivity analysis of our model's calibration error performance concerning two key configurations: the number of sampled answers used to construct the graph and the number of Graph Convolutional Network (GCN) layers in the GNN model. The results are displayed in Fig. 3. The experiments are conducted using the Llama3 model on the TriviaQA dataset. For Fig. 3 (a) experiments, we varied the number of sampled answers from 10 to 50 while keeping other configurations and hyperparameters fixed, as described in the experimental setup. We observe that increasing the number of sampled answers slightly improves performance, which then stabilizes. In Fig. 3(b), the sensitivity to the number of GCN layers indicates that our model remains stable with 1 to 4 layers, with the best performance observed at 3 layers.

## 5 Conclusion and Future Work

In summary, in this work, we proposed an effective strategy of calibrating the confidence an LLM's responses by learning an auxiliary GNN model on the self-consistency pattern among responses to the same questions. Experiments demonstrate that the proposed approach improves confidence calibration significantly across several datasets compared to baseline methods. Our calibration model enhances the reliability of LLMs by evaluating response accuracy, enabling them to abstain from uncertain queries and empowering users to

Table 7: Performance under varying Training Sample Sizes

| Tr. size | ECE | AUROC | Brier |
|---|---|---|---|
| 100 | 0.082 | 0.812 | 0.201 |
| 300 | 0.062 | 0.836 | 0.187 |
| 500 | 0.059 | 0.842 | 0.181 |
| 1000 | 0.047 | 0.853 | 0.177 |
| 4000 | 0.029 | 0.860 | 0.14 |

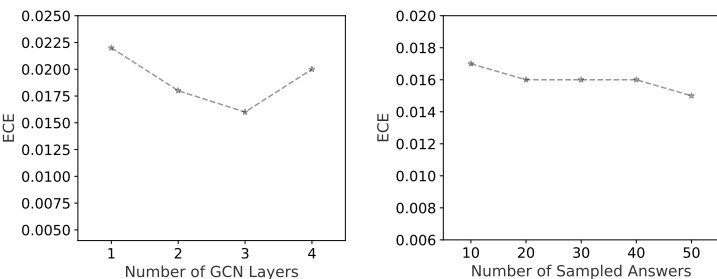

Figure 3: Sensitivity analysis of our model

determine trust levels, thereby promoting responsible deployment in society. However, there are instances where an LLM might be highly confident in an incorrect semantic response, resulting in a consistency graph similar to that of a correct answer. In such cases, our calibration model may not provide an accurate confidence estimation. Without a straightforward solution to this problem, we leave the study of this issue to the future.

**Broader Impacts:** Our work's advancement in calibrating the confidence levels of LLM responses carries important social implications: by accurately estimating the likelihood that a response is correct, an LLM can choose to abstain from answering difficult questions or signal its level of confidence to help users make more informed decisions. We believe this work will enhance the trustworthiness of LLMs and contribute positively to their responsible use in society.

## Acknowledgment

We thank all reviewers and the action editor for their insightful feedback. Li-Liping Liu's work was supported by NSF Award 2239869. We would like to thank Xu Han and Jacob Boerma for their help in labeling the data used in this work.

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

# A  Hyperparameters and Model configurations

**Model hyper-parameters**:

Our model used three GCN(or GIN) layers; typically, the embedding dimension was $256, 512$, and $1024$ for each layers. For the training process, we used the binary cross-entropy loss with a decaying learning rate that reduced the learning rate by 0.9 if the validation loss did not improve 10 epochs (with an initial learning rate of $10^{-4}$ and a minimum learning rate of $10^{-7}$). The optimizer was Adam with $\beta_1 = 0.9$ and $\beta_2 = 0.98$. The batch size was 16, 32. For the rephrased prompts, we set $k = 3$, $n = 30$, so for each rephrased question, we sampled ten answers. While calculating the ECE, we divide the confidence into $B = 10$ bins.

**Evaluation Setup:**

For each question, we evaluate the confidence prediction corresponding to the most likely answers from the LLM response. The setup is consistent with the baseline methods.

**Graph construction:**

For each question, we prompt the LLM to give 30 answers, and the temperature for LLM is set to be 0.6. For each answer, the SentenceBert model Reimers & Gurevych (2019) is used to get each answer's embedding. The cosine similarity between each answer's embedding is taken as the edge weight of the graph. We apply the K-Means clustering method to cluster similar semantic responses. The maximum cluster number is set as 3.

# B  Computational cost

We performed all experiments on NVIDIA A100 GPUs with 80GB of memory. Generating 30 responses using the Llama3 and Vicuna models for 6000 questions from CoQA and TriviaQA data required up to 4 hours, with an average of approximately 2 seconds per question. The CoQA dataset demanded more processing time due to the longer contextual information in the input. The time can be shortened by parallel sampling.

## C   Additional Cases

To better understand our method intuitively, we have collected a few examples to show the difference between our algorithm and APRICOT.

To summarize our observation here:

1. Multiple responses to the same question does reveal the LLM's confidence in its answers. 2. The LLM's self-evaluation of confidence is often much higher than it should be – the LLM is overconfident about its responses. 3. The chain-of-thought responses used by ApriCoT add some information to make each answer more complete and reasonable in the spirit of 1, but it mainly adds the information within one response, not as much information as the multiple responses used by ours.

**Example 1:**

Question: Who plays Captain Jack Sparrow's father Edward Teague in the Pirates of the Caribbean films?

True answer:: Keith Richards

LLM response: David Schofield

More responses from the LLM: Martin Klebba. Keith Richards, Geoffrey Rush, Martin Klebba. Keith Richards. Martin Klebba. David Schofield. (only list 7 responses here to save space)

GCC-estimated confidence: 0.23

CoT response: David Schofield,

Self-evaluation: 80

ApriCoT-estimated confidence: 0.79

**Example 2:**

Question: In which film will you find the Rodger Young?

True answer:: Starship Troopers

LLM response: The Bridge on the River Kwai.

More responses from the LLM: The Greatest Story Ever Told. The Best Years of Our Lives. The Bridge on the River Kwai. The Best Years of Our Lives (1946). 1949's Battleground. The Best Years of Our Lives.

GCC-estimated confidence: 0.22

CoT response: All the President's Men.

Self-evaluation: 95

ApriCoT-estimated confidence: 0.81

**Example 3:**

Question: BS is the international car registration of which country?

True answer:: Bahamas.

LLM response: Germany.

More responses from the LLM: Bahamas. Bahrain. Bangladesh. Bahamas. Belgium. Bahamas. Germany. Bhutan. Belgium.

GCC-estimated confidence: 0.34

CoT response: Belgium

Self-evaluation: 98

ApriCoT-estimated confidence: 0.61

## D   Additional Visualizations

Besides the cases we show in the previous section. Here, we present several case examples and visualize the response patterns. We performed dimension reduction of LLM's responses to different questions and then plotted their embeddings to the 2-dimensional space. Fig 4 shows the responses generated by Llama3 as an example. From the figure, we observe that answers with higher confidence levels tend to cluster closely together, indicating consistency and reliability in these responses. In contrast, answers with lower confidence levels exhibit greater diversity, reflecting a broader range of possibilities. This behavior aligns well with our initial assumption, demonstrating that higher confidence responses are more consistent, while lower confidence responses capture a wider variety of potential answers.

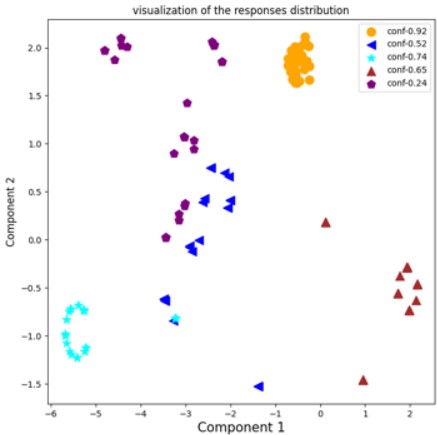

Figure 4: Visualization of the generated response patterns

## E   Additional results

**Additional reliability plots** We showed all reliability diagrams for Llama3 for TriviaQA in Fig. 5 and CoQA dataset in Fig. 6. To summarize the trends, we observe that Platt scaling narrows the range to the middle value. Verbalized uncertainty cannot generate a wider range of confidence values. GraphSpecral with Platt tends to generate a wider range of confidence values, but the bias can not be improved across all cases, resulting in the bar height not following the diagonal line closely. Our model can predict a wider range of confidence values and achieve better calibration in all settings, with the auxiliary consistency graph and clustering features contributing to improved calibration overall.

**Additional baseline results** In Table 8, we showed the performance of the baseline method under varying training sizes. As the number of training data decreases, the ece will drop from 0.096 to 0.165.

Table 8: Performance under varying Training Sample Sizes for the baseline methods(Apricot)

| # of Training Samples | ECE | AUROC | Brier |
|---|---|---|---|
| 100 | 0.165 | 0.611 | 0.229 |
| 300 | 0.133 | 0.634 | 0.211 |
| 500 | 0.112 | 0.695 | 0.204 |
| 1000 | 0.105 | 0.722 | 0.192 |
| 4000 | 0.096 | 0.743 | 0.187 |

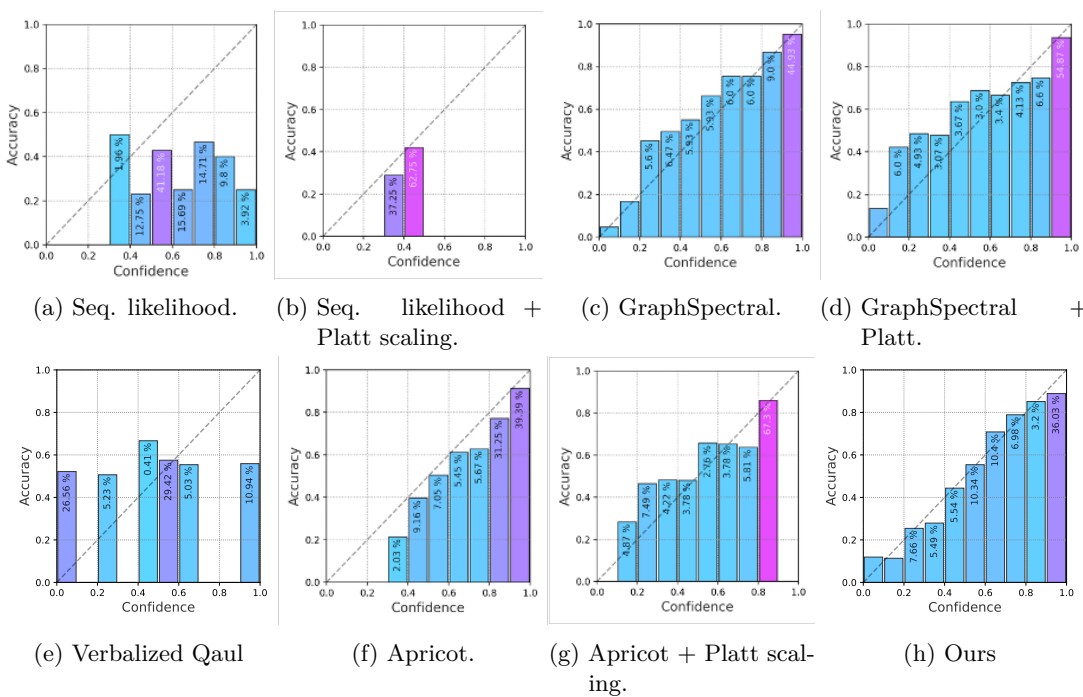

(a) Seq. likelihood.  (b) Seq. likelihood + Platt scaling.  (c) GraphSpectral.  (d) GraphSpectral + Platt.

(e) Verbalized Qaul  (f) Apricot.  (g) Apricot + Platt scaling.  (h) Ours

Figure 5: Reliability diagrams for different methods using 10 bins each for TriviaQA from Llama3 model responses. The color and the percentage number within each bar indicate the ratio of responses contained in each bin. Larger values are represented by colors closer to purple.

## F  Prompting strategy

Here, we showed the prompts to generate the rephrasing questions.

> **Prompts for rephrasing questions**
>
> You are a helpful assistant. I have a question that I would like to see it rephrased in multiple ways. Please take the original question and generate several rephrased versions while maintaining the same meaning, and the question can only have one direct answer. Here is the original question: .... Please provide four distinct rephrases of the question.

The prompts for labeling:

> **Prompts for labeling**
>
> You will be provided with a question, a reference answer, and a student's answer. Please evaluate the student's answer based on the reference answer and provide your score for the student's answer in the format: "Score: ". Assign a score of 0 for incorrect and 1 for correct. For example, "Score: 0" or "Score: 1". Do not include any additional information. Question: {...} Student answer: {...} Reference answer: {...} Now, please enter your score. Score:

## G  Sensitivity and ablations

### G.1  Sensitivity to accuracy metric

In this section, we evaluate the sensitivity of the threshold of accuracy metric of our models. From the results, it show that our method is relative insensitive of the threshold.

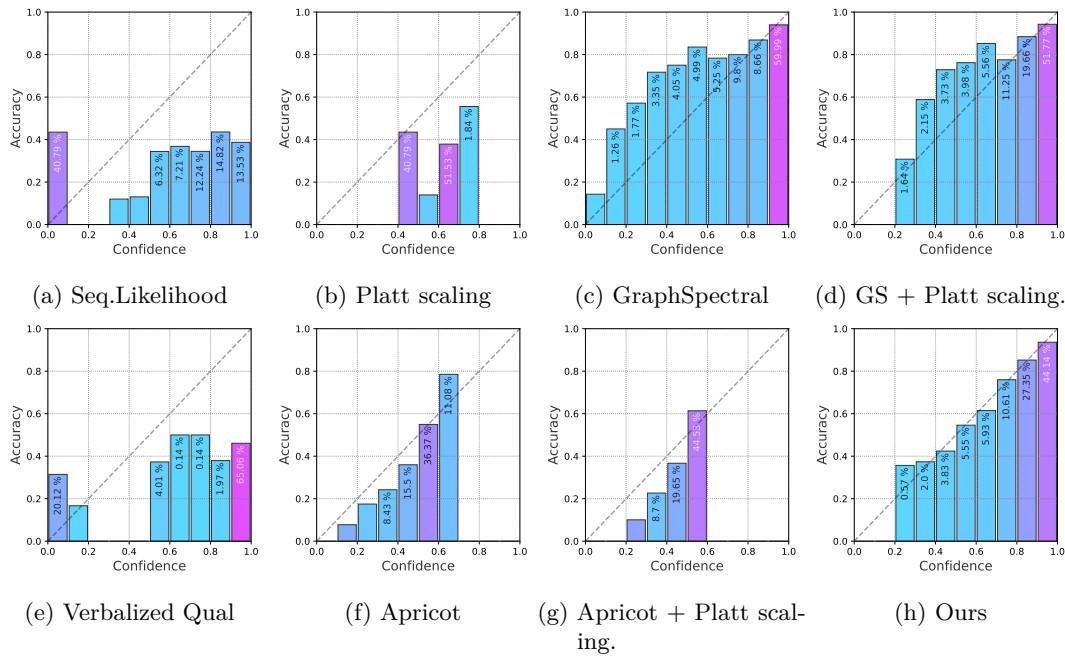

(a) Seq.Likelihood  (b) Platt scaling  (c) GraphSpectral  (d) GS + Platt scaling.

(e) Verbalized Qual  (f) Apricot  (g) Apricot + Platt scaling.  (h) Ours

Figure 6: Reliability diagrams for different methods using 10 bins each for CoQA from Llama3 model responses. The color and the percentage number within each bar indicate the ratio of responses contained in each bin. Larger values are represented by colors closer to purple.

Table 9: The results of using different threshold

| Threshold | AUROC | | | ECE | | |
|---|---|---|---|---|---|---|
| | Apricot | GraphSpectral | Ours | Apricot | GraphSpectral | Ours |
| 0.3 | 0.72 | 0.79 | 0.82 | 0.074 | 0.076 | 0.023 |
| 0.5 | 0.70 | 0.76 | 0.80 | 0.091 | 0.083 | 0.031 |

## G.2 Use the sentence embedding feature as the node feature

We also provide the comparison with using sentence embedding feature as the node feature. We tested this method on TriviaQA. We got Brier scores 0.21, AUROC 0.75, and ECE values 0.11. The results indicate that GNN with sentence embedding as the node feature can produce worse results than our proposed approach. We see clear overfitting issues when GNN uses semantic features: the validation quickly shoots up after the initial dip. We conclude that GNN using semantic features could not generalize to test data.

## G.3 Use the Rouge similarity as the weight of the similarity graph

We provide the results of using Rouge-L as the weight of the similarity graph as shown in Table 10. For the dataset with long-form answers (e.g., TruthfulQA), the performance is much worse than using the clusterID feature, From the results, we conclude Rouge-L is sensitive to the length of the responses.

Table 10: The reuslts of using Rouge similarity graph

| | ECE | AUROC | Brier |
|---|---|---|---|
| TriviaQA | $0.022 \pm 0.006$ | $0.836 \pm 0.000$ | $0.122 \pm 0.000$ |
| CoQA | $0.021 \pm 0.007$ | $0.795 \pm 0.001$ | $0.110 \pm 0.000$ |
| TruthfulQA | $0.035 \pm 0.005$ | $0.632 \pm 0.014$ | $0.221 \pm 0.002$ |

