# OpenReview forum: "Graph-based Confidence Calibration for Large Language Models"
_TMLR — Accepted by TMLR_

### Review · Reviewer_Z1en · 2025-02-09

**Summary Of Contributions:**

The paper presents a method for confidence calibration in large language models by constructing a similarity graph over multiple responses from an LLM to the same question and using a Graph Neural Network to predict correctness probabilities. The authors claim that their approach leverages response consistency to improve confidence estimation outperforms existing methods in calibration metrics such as ECE, Brier Score, and AUROC, and improves out-of-domain generalization compared to prior work.

**Audience:**

Yes

**Claims And Evidence:**

Yes

**Requested Changes:**

## Critical Changes (Required for Acceptance Recommendation):

Clarify the issue of post-hoc correctness labeling. The authors should either use human-annotated correctness labels or show robustness by testing across multiple correctness thresholds. Justify the need for a graph-based approach by providing evidence that a GNN meaningfully improves over alternative consistency-based aggregation methods. Provide an empirical comparison to other confidence calibration methods, such as GraphSpectral, SelfCheckGPT, and APRICOT, and compare performance in both in-domain and out-of-domain settings

## Suggested (Non-Critical) Improvements:

Explain why ROUGE is used instead of SBERT similarity, and provide experimental evidence supporting this choice if it is indeed the best. Discuss broader implications of miscalibrated confidence in LLMs, including real-world risks and failures in domains such as medicine, law, and education. Include a discussion on why previous methods did not fully explore the potential of response consistency and explicitly state what problems their method overcomes that previous approaches failed to address.

**Strengths And Weaknesses:**

## Strengths

The authors conduct comprehensive experiments on multiple datasets, including CoQA, TriviaQA, and TruthfulQA, demonstrating improvements over prior baselines in 3 calibration metrics: ECE, Brier Score, and AUROC. The paper also focuses on out-of-domain generalization, demonstrating greater generalizability of the method in the 3 calibration metrics.

## Weaknesses

The method is too trivial. It simply applies a Graph Neural Network to a dataset that was processed using a non-novel technique. The overall approach lacks originality, as it only takes existing ideas of consistency-based confidence calibration and reformulates them with a GNN. The justification for using GNNs is weak, as the authors do not provide evidence that alternative methods such as clustering or weighted averaging would not work just as well or better. There is no discussion on why a graph representation is fundamentally necessary for this problem, and the claimed benefits are not strongly substantiated.

The correctness labels are post-hoc generated using ROUGE similarity and GPT-4, which introduces bias and allows the authors to tune the label generation process to artificially improve their method’s performance. There is no independent ground truth, making the evaluation unreliable. The choice of ROUGE as a similarity metric is also questionable because it operates at the word level rather than capturing true semantic similarity. The authors do not clarify why why it was chosen over more semantic-aware methods like SBERT similarity.

While the paper compares its approach to some prior consistency-aware methods, such as GraphSpectral and Self-Check GPT, it does not provide a thorough discussion of their limitations. The authors claim that their approach improves upon these methods, but they do not analyze in depth why prior methods fall short in calibration quality or whether the improvements observed in their results are statistically significant.

---

> ### Author Response · Authors · 2025-03-12
> **Rebuttal by authors**
>
> > #### Q1: It simply applies a Graph Neural Network…
> #### Our method differs from Lin et al. in at least the following four aspects. First, we frame it as a learning problem and use a neural network to do the calibration. Second, we consider other measures such as prompt rephrasing to improve the calibration performance. Third, we also investigate our method in the OOD setting. Fourth, we have conducted a more extensive empirical study than Lin et al.. Our method significantly improves over their method in the study.
> #### Methods based on simple data statistics (e.g. GraphSpectral) may not capture the complex relationship between response patterns and confidence levels. Even with simple data statistics, there are still plenty of "parameters" to be decided manually. For example, the choice of similarity metric and the approximation of the number of semantic sets. Manually deciding on these settings may lead to a suboptimal method. As a comparison, our method adaptively learns the pattern and may discover relationships beyond clustering structures. It is also scalable and easy to apply across different datasets and tasks.
>
> > #### Q2: The correctness labels are post-hoc generated using ROUGE similarity…
> #### Previous studies [Kuhn et al.] (table 5) show that ROUGE similarity is accurate for short responses. Following previous work, we use ROUGE on three datasets so we can have a fair comparison against the previous methods. The TruthfulQA dataset has many long answers. We inspected ROUGE and GPT-4 in the label generation step and saw that GPT-4 has a higher accuracy, so we used GPT-4 for this dataset to get response labels, which is also used by previous work [Potsaweel 24].
>
> > #### Q3: While the paper compares its approach to some prior consistency-aware method…
> #### We have added a small paragraph as follows.
>
> #### GraphSpectral and our method have a similar assumption that the consistency level between responses indicates the confidence levels of these responses. However, GraphSpectral uses simple graph statistics to measure the confidence level of responses and could not capture complex relationships between response patterns and confidence levels (e.g. patterns beyond clustering structures). As a comparison, by framing the problem as a learning problem, our method has better opportunities to discover such relationships and provides a better calibration performance. Self-Check GPT uses its own evaluation on whether the context supports the answers and heavily relies on the LLM model’s capability to do self-reflection, which can also be hallucinated. Thus the generated confidence scores are not calibrated well with empirical accuracy.

---

> > ### Author Response · Authors · 2025-03-12
> > **Rebuttal by authors**
> >
> > ### Requested Changes
> > > #### 1. Clarify the issue of post-hoc correctness labeling.
> > #### We tested different thresholds (0.3 and 0.5) over ROUGE scores to obtain labelings of all responses on the TriviaQA dataset. With these labelings, we get their calibration performances as follows:
> > ### Table: Performance of different thresholds over ROUGE scores
> > |Method|AUROC|ECE|
> > |----------|----------|----------|
> > |GraphSpectral|0.79/0.76|0.023/0.031|
> > |Apricot|0.72/0.70|0.074/0.091|
> > |Ours|0.82/0.80|0.023/0.031|
> > #### From these results, we see that our method is insensitive to the correctness thresholds.
> >
> > > #### 2. Justify the need for a graph-based approach…
> > #### We conducted an experiment with the spectral clustering method. We set the number of clusters to be 3. We calculate the cosine distance(see scipy.spatial.distance.cdist) between each pair of responses from different clusters using their embeddings.  We use the average of all these distances as the measure to calibrate the confidence. The rationale is as follows: a large average distance, diverse semantic meanings among semantic clusters, and then low confidence. We run the experiment on the TriviaQA dataset and get the following results:
> > ### Table: Performance on TriviaQA/CoQA using clustering method
> > |Dataset|AUROC|ECE|Brier|
> > |----------|----------|----------|----------|
> > |TriviaQA|0.55|0.26|0.40|
> > |CoQA|0.57|0.19|0.34|
> >
> > #### The performance is much worse than our proposed method. We admit that this is a primitive exploration of this suggested direction. But as we have discussed above, it is non-trivial to develop a good calibration algorithm from simple metrics. For this reason, we would rather make it a learning problem and learn a model to calibrate the confidence.
> >
> > > #### 3.  Provide an empirical comparison to other confidence calibration …
> > #### We have compared GraphSpectral, selfcheckGPT, and ApriCOT on two datasets and two LLM models. We also compare their calibration performance with the expected calibration errors. The empirical evaluation shows that our approach significantly improves the performance since our GNN model learns more useful patterns from the graph structure for confidence calibration and is more robust under out-of-domain situations.
> >
> > ### Suggested Changes
> > > ####  Include a discussion on why previous methods...
> > #### We have added the discussion as below in our manuscript.
> > #### GraphSpectral and our method have a similar assumption that the consistency level between responses indicates the confidence levels of these responses. However, GraphSpectral uses simple graph statistics to measure the confidence level of responses and could not capture complex relationships between response patterns and confidence levels (e.g. patterns beyond clustering structures). As a comparison, by framing the problem as a learning problem, our method has better opportunities to discover such relationships and provides a better calibration performance. Self-Check GPT uses its own evaluation of whether the context supports the answers and heavily relies on the LLM model’s capability to do self-reflection, which can also be hallucinated. Thus the generated confidence scores are not calibrated well with empirical accuracy.

---

### Review · Reviewer_ity6 · 2025-02-21

**Summary Of Contributions:**

The authors introduce a Graph Neural Network (GNN)-based model to estimate response correctness. The key idea is that responses with higher consistency across multiple LLM-generated answers are more likely to be correct. The paper introduces a graph-based method for confidence calibration using response consistency. The authors claim that their approach outperforms existing methods in confidence estimation and OOD generalization.

**Audience:**

Yes

**Claims And Evidence:**

No

**Requested Changes:**

1. Please count how often systemic bias occurs and discuss potential solutions in the paper.
2. Please try to apply this method to complex reasoning tasks/datasets (e.g., MATH, StrategyQA, HotpotQA, etc.)
3. Please try to analyze efficiency trade-offs and explore potential optimizations (e.g., reducing the number of sampled responses).
4. In Tables 1 and 2, please keep the left aligned on the "Method" column to clarify the comparison.
5. At the end of page 8, should "Platting scaling" be "Platt scaling"?

**Strengths And Weaknesses:**

Strengths:
1. The graph-based confidence calibration approach is a novel approach.
2. The proposed method outperforms baseline confidence calibration approaches across multiple datasets (TriviaQA, CoQA, and TruthfulQA).
3. The model holds good calibration performance even in OOD settings.
Weaknesses:
1. The method assumes that consistency among responses correlates with correctness. However, LLMs can be consistently wrong (systemic bias), especially on factual or ambiguous questions (e.g., TruthfulQA). The paper acknowledges this limitation but does not propose a concrete mitigation strategy.
2. The datasets primarily focus on factual QA, but confidence calibration is also crucial in reasoning-intensive tasks (e.g., math problems, multi-hop reasoning).
3. Using ROUGE similarity to assign correctness labels is problematic, as it may not capture deep semantic correctness. You apply GPT-4 to generate labels on TruthfulQA datasets. Why didn't you use it for other datasets?
4. Constructing response graphs for every question requires 30 LLM queries, significantly increasing computational costs.

---

> ### Author Response · Authors · 2025-03-12
> **Rebuttal by authors**
>
> > #### Q1: The method assumes that consistency …
> #### For the issue that an LLM is consistent but wrong, we should consider corrections with extra resources. Actually, our method can help to detect such issues: we can find cases where responses are wrong but have high confidence levels, and then these responses are likely to represent knowledge to be corrected. Therefore, the correctness labels combined with confidence levels can serve as a signal for detecting system biases in an LLM.
>
> > #### Q2&Q6: The datasets primarily focus on factual QA…
> #### We have updated the HopotQA results in the manuscript (last columns of Table 1 & 2). Our proposed method still outperforms baseline methods, which indicates the advantage of using a learning algorithm to do the calibration.
>
> > #### Q3: Using ROUGE similarity to assign correctness labels
> #### Previous studies [Kuhn et al.] (table 5) show that ROUGE similarity is accurate for short responses. Following previous work, we use ROUGE on three datasets to have a fair comparison against the previous methods. The TruthfulQA dataset has many long answers. We inspected ROUGE and GPT-4 in the label generation step and saw that GPT-4 has a higher accuracy, so we used GPT-4 for this dataset.  This method is also used by [Badshah & Sajjad,https://huggingface.co/datasets/potsawee/truthful-qa-llm-judges]. They showed GPT4 level LLMs closely match the human evaluation.
>
> > #### Q4&Q7: Constructing response graphs for every question requires 30 LLM queries…
> #### We have done a study to investigate the relationship between the number of responses and the ECE: the result shows that the ECE only slightly increases as the number of queries decreases. Therefore, it is feasible to use a small number (e.g. less than 10) of queries to calibrate the confidence.
>
> > #### Q5: Please count how often systemic bias occurs and discuss potential solutions in the paper.
> #### On the TriviaQA dataset, we observed a strong statistical correlation between correctness and consistency with R² = 0.966. This suggests that statistically speaking, responses exhibiting greater consistency are more likely to be correct. We further validated this assumption using the TruthfulQA dataset by analyzing the proportion of highly consistent yet incorrect answers. Results showed that only 13% of all incorrect responses exhibited high consistency.
>
> > #### Q8Q9: In Tables 1 and 2, please keep the left aligned on the "Method" column to clarify the comparison.At the end of page 8, should "Platting scaling" be "Platt scaling"?
> #### Yes, we have updated that accordingly.

---

### Review · Reviewer_zZ4u · 2025-02-25

**Summary Of Contributions:**

Confidence estimation for large language models (LLMs) is crucial for ensuring their reliability, especially in real-world applications. Existing methods face challenges in detecting semantic-level mistakes and quantifying uncertainty. This work proposes a novel graph-based confidence calibration method that leverages consistency among an LLM’s multiple responses to a question. A similarity graph is constructed based on the consistency of LLM-generated responses. A Graph Neural Network (GNN) is trained to predict correctness probabilities without processing actual language content. The approach significantly outperforms existing methods across multiple benchmark datasets by using response consistency for confidence estimation.

**Audience:**

Yes

**Broader Impact Concerns:**

There are no broader impact concerns.

**Claims And Evidence:**

No

**Requested Changes:**

There is something off with equation 1.  For the same semantic response C, there can be numerous r when we consider the domain of natural language responses.  Consequently, p(C|q) = \Sum_r∈C p(r|q) can lead the p(C|q)  to be greater than 1 if p(r | q) is estimated using something like softmax. I think there is some assumption here that the space of all possible responses is somehow known and can be enumerated. For natural language responses without some syntactic restriction (such as the count of tokens), one can't make such an assumption.

Using ROUGE similarity to compute semantic similarity is odd. It would be useful to consider some other alternatives such as embedding in eqn 4 and empirically evaluate what measure is a good choice.

Similarly, for equation 6 in method 2, why not also consider ROGUE metric when measuring similarity between r_i and r_j?

"We choose NOT to feed in the embedding vectors of responses to avoid the GNN’s dependency on textual information" - once again, this choice must be studied through ablation. Just relying on how the responses are split across clusters using 1-hot embedding would remove significant semantic content and rely on the distribution of semantic dispersion of the output to help predict confidence.

" We employ a multiple rephrased questions strategy for answer sampling. Specifically, we prompt the GPT-4 to give k different but with the same meaning rephrased questions for the given question q. Then, we sample n/k responses from the LLM for each rephrased question and still get a total of n responses" - how sensitive is the method to the use of different LLMs for this prompt rewriting?

Overall, the method presented in the paper appears rather simplistic and an incremental extension of Lin et al 2024. Without sufficient empirical study, including the suggestions above, it is not clear if the claims in the paper are well-justified.

**Strengths And Weaknesses:**

+ The paper attempts to address a very important problem of confidence calibration of LLMs

---

> ### Author Response · Authors · 2025-03-12
> **Rebuttal by authors**
>
> > #### Q1: There is something off with equation 1…
> #### Here r represents a sequence as the response to q, so p(r|q) is the probability of the sequence. The probability can be decomposed as the probability of individual tokens given previous tokens in an LLM, so p(C|q) will not be greater than 1. In our analysis, we don't really enumerate all possible r, but only use it as a tool to justify the approximate probability p(C | q) of a semantic class C. Our analysis is similar to that of Kuhn et al. .
>
> > #### Q2: Using ROUGE similarity…
> #### Obtaining True/False labels of LLM's responses from their similarities to the true response is an important step in the confidence calibration task. We have followed the convention in the field and used ROUGE as the similarity metric. Previous studies by Kuhn et al. (Table 5) show the labeling method with ROUGE closely matches the human evaluation, especially when responses are relatively short.
> #### We have to use ROUGE in the labeling step in order to compare our method with previous methods in an equal way. Within our method, we can use ROUGE to construct a graph and evaluate confidence values. The results are
> ### Table :Performance of using ROUGE similarity to construct a graph
> |Dataset|AUROC|ECE|Brier|
> |----------|----------|----------|----------|
> |TriviaQA|0.83 &plusmn; 0.000|0.022 &plusmn; 0.006 |0.12 &plusmn; 0.000|
> |CoQA|0.79 &plusmn; 0.001|0.021 &plusmn; 0.007|0.11 &plusmn; 0.000|
> |TruthfulQA|0.63 &plusmn; 0.014 |0.035 &plusmn; 0.005|0.22 &plusmn; 0.002|
>
>
> > #### Q3:  "We choose NOT to feed in the embedding vectors…
> #### We conducted the ablation study as suggested. Instead of using node features extracted from the graph, we feed in the embedding vectors of responses as node features of the graph. We tested this method on TriviaQA datasets and got Brier scores 0.21, AUROC 0.75, and ECE values 0.11, which are much worse than our proposed approach. We see clear overfitting issues when GNN uses semantic features: the validation quickly shoots up after the initial dip. We conclude that GNN using semantic features could not generalize to test data.
>
> > #### Q4: We employ a multiple-rephrased…
> ####  With the rephrasing, we have observed relatively higher performance across the board, but the improvement is small, so we think the method is not very sensitive to prompt rewriting.
>
> > #### Overall, the method presented in the paper appears
> ####  Our method differs from Lin et al. in at least the following four aspects. First, we frame it as a learning problem and use a neural network to do the calibration. Second, we consider other measures such as prompt rephrasing, which leads to small but consistent improvements in the calibration performance. Third, we also investigate our method in the OOD setting. Fourth, we have conducted a more extensive empirical study than Lin et al.. Our method significantly improves over their method in the study.

---

### Review · Reviewer_EEyf · 2025-02-25

**Summary Of Contributions:**

This paper presents a method for estimating the confidence of an LLM generated answer. The approach measures the pair-wise similarity among multiple LLM generated responses, and uses the resulting similarity graph as the input to a trained graph neural network to estimate probability of a correct response.  In experiments on multiple benchmarks, the proposed method shows improved results with regards to calibration accuracy, including in OOD settings. The paper also presents sensitivity analyses, exploring the impact of training data size and hyperparameter sensitivity.

**Audience:**

Yes

**Broader Impact Concerns:**

The paper might be improved with a brief broader impact statement describing the risks of overreliance on AI models. Confidence estimates can help people avoid overreliance, but only if they are correct, and that is worth noting in the context of a broader impact.

**Claims And Evidence:**

Yes

**Requested Changes:**

- add statistical significance to experimental results
- add a sensitivity analysis over rephrasing methods

**Strengths And Weaknesses:**

Strengths:
* the paper is well written and organized.
* the experiments show improvements in confidence calibration.

Weaknesses:
- statistical significance of improvements is unclear.

Questions:
- which of the results in table 1 and table 2 are statistically significantly better than baselines?

- the work is presented as providing an estimate of correctness.  However --- as acknowledged in the conclusion --- there are many reasons why the LLM might be confident but wrong, such as bad or wrong training data or a significant distribution shift in the model prompt.  It would be useful to find an alternative phrase early in the paper, perhaps something like a confidence estimate "regarding the model alignment (with the training data)" or "reliability relative to the training data".

- a sensitivity analysis over rephrasings of "the original question q in k different forms" would be useful.  How strongly does the effectiveness of the method depend on the approach?

- Lin et al. (2024)'s GraphSpectral approach, as described in the related work sounds similar to this proposal.  What are the graph statistics used by them and how do the learned GNN's responses improve on it?

---

> ### Author Response · Authors · 2025-03-12
> **Rebuttal by authors**
>
> > Q1: Which of the results in Table 1 and Table 2 are statistically significantly better than baselines?
>
> #### We have added the statistical significance of the results in the manuscript. We repeat the experiments 10 times, each time with a different train/validation split and test the performance on the test set.
>
> > Q2: The work is presented as providing an estimate of correctness. … "reliability relative to the training data".
>
> #### We have modified the introduction section to a more accurate phrase:
> #### Our work focuses on common real-world scenarios, assessing how well the LLMs align with the knowledge of the training data. This work does not consider the case where the training data contains consistent but wrong knowledge.
>
> > #### Q3: a sensitivity analysis over rephrasings ... How strongly does the effectiveness of the method depend on the approach?
> #### Including different forms of the same query improves the performance across the board, but the improvement is minimal.
>
> > Q4: Lin et al. (2024)'s GraphSpectral approach … how do the learned GNN's responses improve on it?
>
> #### The major difference between Lin et al. (2024) and our work is that we frame the calibration problem as a learning problem.  Lin et al. use manually designed graph statistics to rank confidence levels of responses to different questions and do not provide a calibration.
> #### Graph Spectral uses the "sum of eigenvalues of the graph Laplacian", which captures cluster patterns among responses, to measure the confidence levels of responses. The simple statistics may not capture the complex relationship between responses and confidence levels.   In contrast, our method takes a learning approach and thus has a stronger ability to discover patterns for confidence estimation. This hypothesis is verified in our extensive experiments, which shows that our method has lower calibration error than the GraphSpectral baseline and other baselines.

---

### Comment · Action_Editor_ffTv · 2025-03-12

Dear authors,

The discussion period has ended and it appears you haven't submitted yet any responses to the posted reviews. I encourage you to still submit a reply at your earliest convenience.

---

> ### Author Response · Authors · 2025-03-12
> **Thanks for all reviewers!**
>
> Dear Editor and Reviewers,
>
> Thank you again for your valuable feedback. We have carefully revised our work to address the reviewers’ concerns:
> 1. We have added additional experimental results and conducted a thorough sensitivity analysis.
> 2. We have enhanced the discussion by providing a more comprehensive comparison between our approach and previous methods.
> 3. We have refined the introduction and experiment section to improve clarity and precision.
>
> Below are the detailed responses for each reviewer.

---

### Comment · Action_Editor_ffTv · 2025-04-08

Dear authors,

Thank you for your submission and the interesting paper. Multiple reviewers have recognized the importance of the topic and the values of the proposed approach. Several reviewers have independently brought up the following two main concerns:

1. Relation to prior work (e.g. Lin et al. 2024): While technique novelty is not a acceptance requirement to TMLR, since the paper itself is making novelty claims in the abstract etc., there is a need to convincingly justify them. Multiple reviewers have reiterated that this concern remains also after the rebuttal responses. That said, there have been little doubt about the interest of the community in the paper. I recommend to either convincingly justify such claims, or to adjust the presentation.

2. Correctness metrics: Multiple reviewers expressed concerns about only using Rouge to measure whether an answer is correct or not. The authors response during the rebuttal claimed that prior work also relied on it. However, (a) previous work relied more on auto eval models and also (b) performed human verification to assess reliance. And (c) the learning nature of this work raised more concerns with this practice due to higher chance of learning biases in the measuring metric.

Would it be possible to address these two points?

Thank you very much

---

### Decision · Action_Editor_ffTv · 2025-04-21

**Recommendation:** Accept with minor revision

**Comment:**

This paper introduces a GNN based method for calibrating the confidence of LLMs on top of multiple resampled answers, with the goal of  improving the uncertainty estimation for the correctness of the selected answer. The work builds on previous self-consistency papers such as  (Chen & Mueller, 2023; Lin et al., 2024) and (Ulmer et al., 2024) and suggests to add a GNN for improving the confidence prediction. Overall, this is a focused study that introduces a new method and empirically evaluates it across in-domain and out-of-domain datasets with two LLMs. It will be interesting to expand this direction further with long output answers and more capable LLMs.

The final paper should include all the additional results and clarifications discussed from the discussions below.

**Audience:**

Yes. LLM confidence topic is of high interest to TMLR audience.

**Claims And Evidence:**

This paper introduces a GNN based method for calibrating the confidence of LLMs on top of multiple resampled answers, with the goal of  improving the uncertainty estimation for the correctness of the selected answer. The main claims are that the proposed GNN method improves calibration on both in-domain and out-of-domain setups. This is evaluated using two LLMs (Vicuna and LLama3) on several short answer output datasets in a cross-validation setup, and using standard ECE/ Brier/ AUC metrics. Results show that the proposed method performs better on these metrics.

For evaluating whether the answer is correct or not, a threshold over ROUGE-L similarity with the gold answer was used. Reviewers brought up valid concerns about this practice, especially since the training of the GNN can catch biases in the ROUGE metric. The authors have added results with manual annotations for a subset of the datasets.

Another common concern by reviewers was the positioning of the paper compared to previous work such as Lin et al. 2024. The authors have revised the paper to more clearly position the contribution around improving the calibration with the method and conducting empirical studies.

---

> ### Author Response · Authors · 2025-04-24
>
> We sincerely thank the reviewers and the action editor for their thoughtful and constructive feedback, which greatly improved the clarity and quality of our manuscript. We will make further changes and post the camera-ready version soon.

---

> > ### Author Response · Authors · 2025-05-11
> > **The camera-ready version is uploaded**
> >
> > Dear action editor,
> >
> > We have submitted the camera-ready version. Here are some further changes.
> >
> > 1. We have added the experiments with manual labels as a subsection of the work, considering the importance of the quality of correctness labels.
> >
> > 2. We have added a paragraph discussing the broader impact of this work.
> >
> > 3. We have gone through a thorough editing pass, which has fixed small writing issues and improved the arrangement of tables and figures.
> >
> > Please let us know if there are further issues.
> >
> > -- Paper4032 authors